# Visible Yet Unreadable: A Systematic Blind Spot of Vision–Language Models Across Writing Systems

**AI**
OpenAI, Anthropic, and Google

**Jie Zhang**[*]
CFAR and IHPC, A*STAR

**Ting Xu**
National University of Singapore

**Gelei Deng**
Nanyang Technological University

**Runyi Hu**
Nanyang Technological University

**Han Qiu**
Tsinghua University

**Tianwei Zhang**
Nanyang Technological University

**Qing Guo**[†]
Nankai University

**Ivor Tsang**
CFAR and IHPC, A*STAR

## Abstract

Writing is a universal cultural technology that reuses vision for symbolic communication. Humans display striking resilience: we readily recognize words even when characters are fragmented, fused, or partially occluded. This paper investigates whether state-of-the-art vision–language models (VLMs) share this resilience. We construct two psychophysics-inspired benchmarks across distinct writing systems—Chinese logographs and English alphabetic words—by splicing, recombining, and overlaying glyphs to yield "visible-but-unreadable" stimuli for models while remaining legible to humans. Despite strong performance on clean text, contemporary VLMs show a severe drop under these perturbations, frequently producing unrelated or incoherent outputs. The pattern suggests a structural limitation: models heavily leverage generic visual invariances but under-rely on compositional priors needed for robust literacy. We release stimuli generation code, prompts, and evaluation protocols to facilitate transparent replication and follow-up work. Our findings motivate architectures and training strategies that encode symbol segmentation, composition, and binding across scripts, and they delineate concrete challenges for deploying multimodal systems in education, accessibility, cultural heritage, and security.

## 1 Introduction

Reading is one of humanity's most powerful cultural inventions [1, 2]. By mapping arbitrary visual marks onto symbolic meaning, writing systems enable the storage, transmission, and accumulation of knowledge across generations. Despite their vast diversity—from logographic Chinese characters to alphabetic scripts—humans exhibit striking resilience in reading: we readily recognize text even when it is cut, overlapped, occluded, or distorted [3, 4]. This robustness likely reflects deep structural priors in human perception—the expectation that written symbols are composed of parts and follow compositional rules that support recovery from incomplete input [5].

Artificial intelligence, by contrast, often appears to read but lacks the same resilience. Vision–language models (VLMs) can transcribe rendered text, answer questions about documents, and

---

[*]https://zjzac.github.io/
[†]The corresponding author

interleave language with images [6, 7, 8]. Yet their reading ability has not been systematically probed under distortions that are trivial for humans. We ask a simple but fundamental question: *Can models read what humans can still read?* Our results indicate that, despite near-perfect performance on clean text, state-of-the-art VLMs show a *sharp degradation* when confronted with perturbed but human-readable constructs, revealing a robust cross-script gap between human and machine literacy.

To probe this gap, we design psychophysics-inspired benchmarks across two distinct writing systems. In Chinese logographs, we render a set of 100 four-character idioms and splice each character along horizontal, vertical, or diagonal axes, recombining fragments into composite glyphs. In English alphabetic words, we select 100 eight-letter words, divide them into two halves, render each half in different colors, and overlay them to form fused stimuli. These manipulations preserve human legibility while consistently reducing VLM accuracy, with models frequently producing unrelated or incoherent outputs. The observed pattern is consistent with a structural limitation: generic visual invariances learned from large-scale training [9, 10] may be insufficient for textual identifiability without stronger symbol-centric priors.

The implications extend beyond technical benchmarks. Robust machine literacy under mild perturbations matters for scientific curation of handwritten notes and historical manuscripts [11, 12], for education and accessibility in non-standard scripts and diverse reader populations [13], for cross-lingual and low-resource contexts [14], and for security-sensitive document analysis where adversaries may exploit brittleness [15]. More broadly, gaps between human and machine reading bear on the reliability of multimodal AI systems [16, 17].

**Contributions.** This study makes four contributions:

- **Evidence of a cross-script failure mode:** We show that state-of-the-art VLMs exhibit a marked collapse on "visible-but-unreadable" stimuli—perturbations that remain trivial for humans across logographic and alphabetic scripts.
- **Psychophysics-inspired benchmarks:** We introduce controlled perturbations for Chinese idioms and English words that preserve human readability while systematically challenging machine recognition.
- **Implications for model design:** Findings highlight a fundamental human–AI asymmetry in literacy and suggest that scaling alone may be insufficient, motivating architectures with explicit structural priors for segmentation, composition, and binding [18, 19].
- **Broader impact:** We delineate risks and opportunities for deploying robust machine reading in security, scientific curation, education, accessibility, cultural heritage, and trustworthy multimodal AI.

## 2   Related Work

### 2.1   Human Reading, Psychophysics, and Structural Priors

Human readers exhibit remarkable robustness to distortions in writing. Classic psychophysics shows that crowding, occlusion, and fragmentation often impair object recognition, yet humans can still recover meaning from incomplete text [20, 21]. This resilience has been linked to structural priors such as segmentation, binding, and morpheme- or radical-level expectations [5]. Our work situates itself in this tradition: we show that when structural boundaries of glyphs are disrupted, humans maintain readability but vision–language models (VLMs) collapse, exposing the absence of such priors in current architectures.

### 2.2   Multimodal VLMs and Reading Ability

Large VLMs such as CLIP [22], BLIP-2 [23], Kosmos [24], LLaVA [8], GPT-4V, and Gemini have demonstrated impressive capabilities in multimodal understanding, including text-in-image tasks. Evaluations on document QA, chart understanding, and OCR-VQA benchmarks suggest that these models can "read." However, these evaluations overwhelmingly use natural renderings of text. Our results reveal that once structural perturbations are introduced, VLMs fail systematically—even when humans find the text trivial to recognize. This highlights that current VLM "reading" is more a byproduct of visual invariance than of symbolic identifiability.

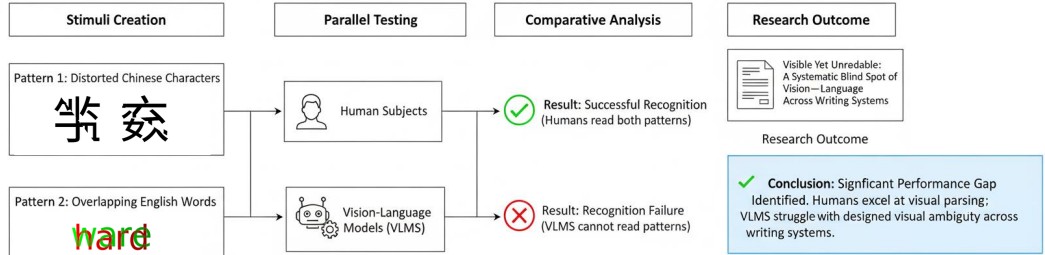

Figure 1: Overview of the experimental framework. Distorted Chinese characters and overlapping English words were created as stimuli and tested in parallel on human subjects and vision–language models (VLMs).

## 2.3 Psychophysics-inspired Evaluation in Machine Learning

Recent work in machine vision has drawn from psychophysics, using controlled parametric stimuli to generate accuracy–perturbation curves [25, 26]. Such methods have probed frequency sensitivities, crowding effects, and texture–shape biases. We extend this methodology to the domain of reading, introducing controlled perturbations that sever the link between visibility and identifiability. Our cross-script benchmarks expose the tension between generic invariance and symbolic recognition more sharply than prior visual metamer or distortion studies.

## 2.4 Sub-character Visual Information in Chinese NLP

Complementary to our findings, recent work has examined whether large language and vision–language models can recognize and exploit the sub-character structure of Chinese writing. For instance, Wu *et al.* [27] construct a benchmark to evaluate models' ability to leverage radicals, strokes, and compositional structure in Chinese characters. Their results show that models exhibit a limited sensitivity to these visual features, and that explicitly providing radical information in prompts can improve downstream tasks such as part-of-speech tagging. In contrast, our study does not probe whether models can partially utilize radicals, but instead demonstrates a systematic failure when characters or words are perturbed in ways that remain fully legible to humans. Whereas [27] highlight the potential for models to benefit from sub-character information in Chinese, we highlight a broader architectural blind spot across both logographic and alphabetic scripts, showing that scaling and prompting are insufficient to bridge the human–machine gap in robust reading.

# 3 Methods

## 3.1 Overview

Our goal is to test whether state-of-the-art vision–language models (VLMs) can read stimuli that remain legible to humans but violate structural assumptions of conventional text. We design two psychophysics-inspired benchmarks—one based on Chinese logographs and the other on English alphabetic words—and evaluate multiple VLMs under controlled perturbations. Figure 1 illustrates the overall pipeline.

## 3.2 Stimuli Construction

**Chinese logographs.** We construct a dataset of 100 four-character idioms (*chengyu*), a canonical and semantically coherent unit in Chinese. Each character image is rendered in a standard font at fixed resolution. As shown in Figure 2, to generate perturbed stimuli, we apply one of three cutting operations:

- **Horizontal cut:** split the glyph into upper and lower halves, then recombine mismatched parts.
- **Vertical cut:** split the glyph into left and right halves, recombining across characters.
- **Diagonal cut:** split along the main diagonal, recombining triangular fragments.

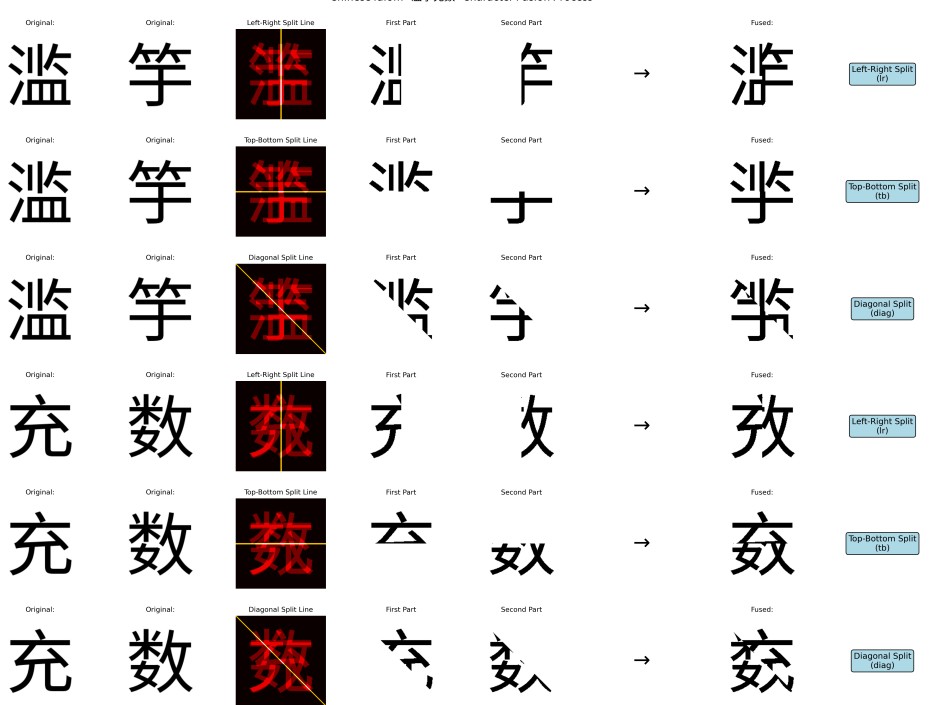

Figure 2: Illustration of the character fusion process for the Chinese idiom "滥竽充数". Each pair of original characters is split along different axes (left–right, top–bottom, or diagonal) and recombined into fused forms. The procedure demonstrates systematic ways of generating visually ambiguous but linguistically interpretable characters, which serve as stimuli for evaluating recognition robustness.

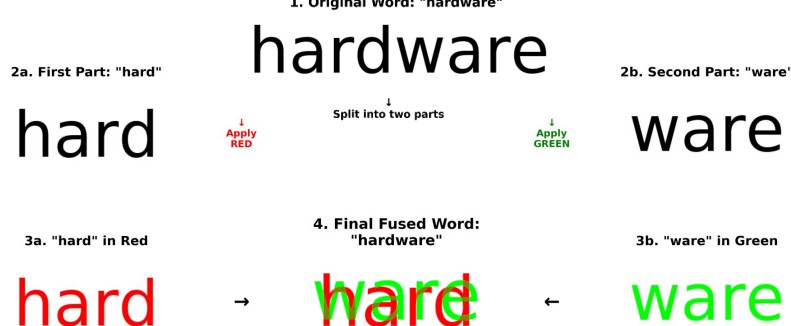

Figure 3: Illustration of the English word fusion process using "hardware" as an example. The original word is split into two segments ("hard" and "ware"), which are then color-coded (red and green) and recombined into an overlapping fused form.

These operations preserve local stroke visibility but destroy coherent glyph boundaries. The resulting composites remain readable to human subjects, who can reconstruct the intended idiom with near-perfect accuracy, but appear as novel glyph-like objects to VLMs.

**English alphabetic words.** We sample 100 eight-letter words from a standard English lexicon. As shown in Figure 3, taking "hardware" as an example, each word is split into two four-letter halves. The first half is rendered in red, the second half in green, with consistent font and resolution. The two halves are then overlaid to form a single composite image. Humans can reliably parse the superimposed text, but the overlapping colors and fused boundaries challenge models that lack explicit segmentation mechanisms.

Table 1: Prompt design for the Chinese idiom fusion task.

| Type | Prompt | English Translation |
|---|---|---|
| Basic | 请识别这张图片中的汉字，直接返回识别出的文字，不需要其他解释。 | Identify the Chinese characters in the image and return only the recognized text, without any explanation. |
| Detailed | 这是一张包含合体字的图片，每个字由两个汉字的部分组合而成。请仔细观察并识别出原始的汉字内容。只返回识别出的文字。 | This image contains fused Chinese characters, where each character is composed of parts from two different characters. Carefully identify the original characters and return only the recognized text. |
| Contextual | 这张图片显示的是中文成语的艺术字体，其中每个字都是由两个字的部分拼接而成。请识别出这个四字成语。 | This image shows an artistic rendering of a Chinese idiom, where each character is formed by fusing parts of two characters. Identify the complete four-character idiom. |

Table 2: Prompt design for the English word fusion task.

| Type | Prompt |
|---|---|
| Basic | What text do you see in this image? |
| Detailed | This is an 8-letter English word where each half has been diagonally fused together. What word is it? |

## 3.3 Models Evaluated

We evaluate a range of widely used VLMs, including open-source models (e.g., Qwen2-VL-7B, [28] LLaVA-Mistral-7B [8], and LLaVA-Next-Vicuna-7B [29]) and proprietary frontier models (e.g., OpenAI GPT-4o [30], GPT-5 [31], Anthropic Claude Opus 4.1 [32], Sonnet 4 [33], Google Gemini 1.5 Pro and 1.5 Flash [34]). All models are accessed through their publicly available APIs or checkpoints, without any additional fine-tuning.

For the Chinese idiom fusion task, we design three prompt settings (see Table 1). For the English word fusion task, we design two prompt settings (see Table 2). To improve reliability and reduce unnecessary token usage in English, we further adopt a **concise prompting strategy**, appending *"Answer with just the word, no explanation or thinking."* to the end of each prompt.

## 3.4 Evaluation Protocol

For each stimulus, we collect the model's textual output and compare it to the ground truth. Accuracy is measured differently for Chinese idioms and English words:

- **Chinese idioms:** We adopt two complementary evaluation metrics, both applied after preprocessing by removing all non-Chinese characters using the regex.
  - **Strict Match:** A Boolean indicator of whether the prediction is identical to the ground truth.
    * *Definition:* Correct if and only if `prediction == ground_truth`.
    * *Implementation:* Direct string comparison.
    * *Usage:* Used for computing idiom-level accuracy.
  - **Average Similarity:** A soft metric that measures partial correctness.
    * *Computation:* Sequence similarity is computed via `difflib.SequenceMatcher`.
    * *Return value:* A floating-point score between 0 and 1, where 1 indicates a perfect match.
- **English words:** We only evaluate at the whole-word level, using *Exact Match* accuracy (i.e., the predicted word must match the ground truth exactly).
- **Human baseline:** We recruit 10 participants (native speakers for the corresponding script) and measure accuracy under the same stimuli. We randomize order, enforce attention checks, and report aggregate accuracy with confidence intervals, as well as per-item confusion patterns.

Table 3: Recognition accuracy of idiom images with character-splitting rendering under three prompting strategies. The table reports strict matching rate (exact idiom match), average matching rate (similarity-based match), and human evaluation results (all rated as 100%).

| Model | Strict Matching Rate | Average Matching Rate | Human Evaluation |
|---|---|---|---|
| **OpenAI** | | | |
| gpt-4o/basic | 0.0% | 11.1% | 100% |
| gpt-4o/context | 0.0% | 5.2% | 100% |
| gpt-4o/detailed | 0.7% | 7.7% | 100% |
| gpt-5/basic | 0.0% | 10.5% | 100% |
| gpt-5/context | 5.0% | 11.4% | 100% |
| gpt-5/detailed | 1.5% | 12.1% | 100% |
| **Anthropic** | | | |
| claude-opus-4-1/basic | 1.0% | 15.5% | 100% |
| claude-opus-4-1/context | 0.0% | 1.8% | 100% |
| claude-opus-4-1/detailed | 5.2% | 14.7% | 100% |
| claude-sonnet-4/basic | 0.0% | 10.1% | 100% |
| claude-sonnet-4/context | 0.0% | 1.4% | 100% |
| claude-sonnet-4/detailed | 0.8% | 7.3% | 100% |
| **Gemini** | | | |
| gemini-1.5-flash/basic | 0.3% | 7.5% | 100% |
| gemini-1.5-flash/context | 5.0% | 10.6% | 100% |
| gemini-1.5-flash/detailed | 1.5% | 11.0% | 100% |
| gemini-1.5-pro/basic | 0.0% | 8.0% | 100% |
| gemini-1.5-pro/context | 0.2% | 3.4% | 100% |
| gemini-1.5-pro/detailed | 0.3% | 7.5% | 100% |
| **LLaVA** | | | |
| llava-mistral-7b/basic | 0.0% | 0.6% | 100% |
| llava-mistral-7b/context | 0.0% | 0.5% | 100% |
| llava-mistral-7b/detailed | 0.0% | 0.6% | 100% |
| llava-next-vicuna-7b/basic | 0.0% | 0.6% | 100% |
| llava-next-vicuna-7b/context | 0.0% | 0.5% | 100% |
| llava-next-vicuna-7b/detailed | 0.0% | 0.6% | 100% |
| **Qwen** | | | |
| qwen2-vl-7b/basic | 0.0% | 24.4% | 100% |
| qwen2-vl-7b/context | 0.0% | 13.9% | 100% |
| qwen2-vl-7b/detailed | 0.0% | 24.0% | 100% |

# 4 Experiments

## 4.1 Overall Results

Across both the Chinese idiom and English word fusion tasks, all evaluated VLMs show a substantial performance gap compared to human recognition (100% across all cases).

For the Chinese idiom task (Table 3), strict matching accuracy (exact idiom recognition) remains extremely low across all models, typically below 5%. Even with similarity-based evaluation, average matching rates rarely exceed 15%, with the exception of Qwen2-VL-7B, which achieves around 24%. Prompt design has only limited impact: detailed prompts provide modest improvements for some models (e.g., GPT-5, Claude Opus 4.1), while context-oriented prompts do not consistently help.

For the English word fusion task (Figure 4), overall performance is likewise poor, with recognition accuracy capped at 20% even under detailed prompts (GPT-5). Proprietary frontier models (e.g., GPT-4o, GPT-5, Gemini 1.5) show slightly better performance than open-source alternatives (LLaVA, Qwen2-VL), yet still fall far below human-level recognition. Detailed prompts consistently outperform basic prompts, indicating that explicit task guidance can mitigate, but not resolve, the recognition challenge.

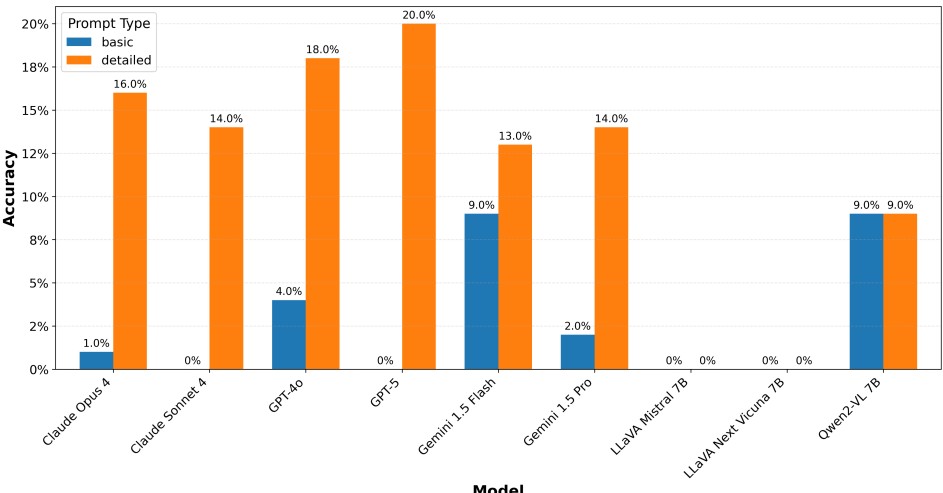

Figure 4: Performance of vision–language models (VLMs) on the English word fusion recognition task. Accuracy is reported under two prompting conditions (basic vs. detailed). Overall performance is low across all models, with proprietary frontier models (e.g., GPT-4o, GPT-5, Gemini) achieving up to 20% accuracy under detailed prompts, while most open-source models (LLaVA variants, Qwen2-VL) perform near chance level.

Figure 5: Word recognition difficulty comparison. Top: hardest words (0% recognition rate). Bottom: easiest words (16.7%-55.6% recognition rate).

Taken together, these results highlight a systematic blind spot in current VLMs: while humans easily parse visually fused characters and words, models fail almost completely, regardless of architecture, scale, or prompting strategy.

## 4.2 Recognition Difficulty Analysis

To better understand model limitations, we further analyze recognition difficulty across individual stimuli (Figures 5, 6).

For the **English word fusion task** (Figure 5), recognition rates vary widely across words. The hardest cases, such as *hardware*, *checksum*, and *decoding*, exhibit complete failure across models (0% recognition). These words involve highly overlapping letter structures or stroke collisions, which appear to overwhelm the visual parsing mechanisms of current VLMs. In contrast, relatively easier words such as *keyboard*, *alphabet*, and *password* achieve recognition rates between 38.9% and 55.6%, suggesting that clearer segmentation cues or less stroke overlap can partially mitigate the ambiguity.

For the **Chinese idiom fusion task** (Figure 6), the difficulty gap is even more pronounced. Many idioms are never correctly recognized (0%). Even the relatively "easiest" idioms only reach 2.5%–8.0%

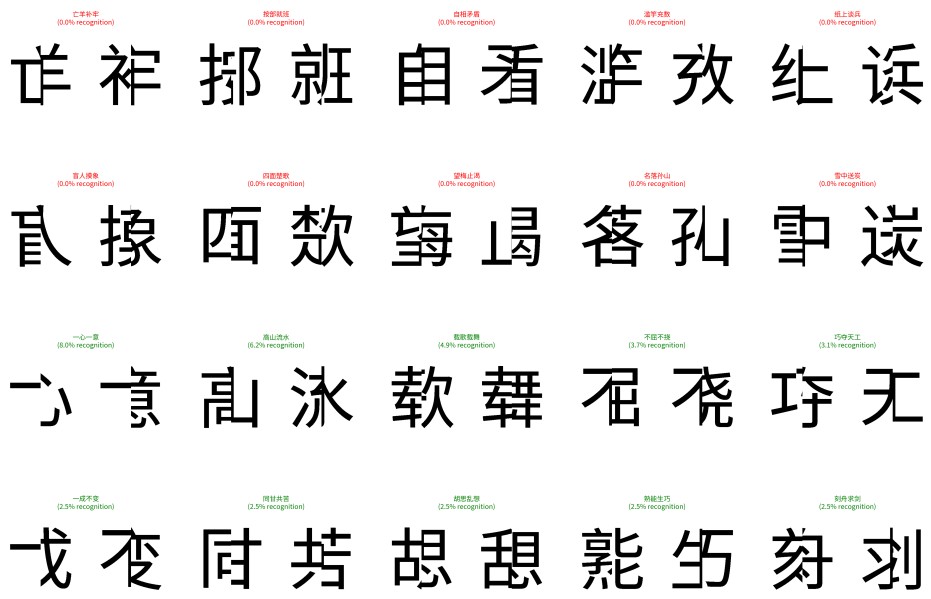

Figure 6: Idioms recognition difficulty comparison. Top: hardest idioms (0% recognition rate). Bottom: easiest idioms (2.5%-8% recognition rate).

accuracy. The logographic structure of Chinese characters amplifies this problem: fusing parts of two characters often produces a visually valid but semantically misleading glyph, making it especially challenging for VLMs.

Importantly, however, this gradation of difficulty is *not reflected in human perception*. Human participants reported no meaningful difference between "hard" and "easy" examples, recognizing all items near-perfectly regardless of stroke overlap or fusion style. This divergence highlights that the observed error spectrum is not intrinsic to the stimuli themselves, but rather emerges from architectural blind spots in current VLMs. Whereas humans rely on robust gestalt grouping, contextual priors, and flexible character reconstruction, VLMs lack mechanisms to resolve structured visual ambiguity, leading to stark performance discrepancies that do not exist in human cognition.

In a nutshell, these results reveal that difficulty for VLMs is an artifact of model limitations rather than an inherent property of the task. This underscores a systematic blind spot in visual–linguistic parsing, spanning both alphabetic and logographic writing systems.

## 5   Discussion and Conclusion

Our study reveals a universal failure mode in vision–language models (VLMs): accuracy that is near-perfect on clean text collapses to near-zero under perturbations that remain fully legible to humans. This gap highlights a fundamental cognitive asymmetry. Humans read by deploying structural priors—segmentation, composition, and symbol binding—while VLMs rely on global invariances that misfire when identifiability is challenged.

The implications are profound. Reading is not mere pattern recognition but structured symbol recovery. Humans achieve this effortlessly; VLMs do not. While larger models and more data may help, our results suggest that scaling alone may be insufficient. Instead, models must incorporate literacy-oriented priors: glyph- or radical-aware representations, mechanisms for segmentation and binding, and cross-script strategies that generalize across writing systems.

The "visible-but-unreadable" blind spot matters well beyond safety. Robust reading under perturbation is essential for science, accessibility, cultural heritage, and trustworthy multimodal AI. Addressing this gap is therefore not a minor refinement but a prerequisite for building AI that can partner with humans in domains where literacy is indispensable.

Our study is limited to two scripts and controlled perturbations. Expanding to more languages, fonts, and distortions, combined with systematic human studies, will help map the full scope of the problem and inspire design. Exploring symbolic-neural hybrids and explicit segmentation architectures remains an open and promising frontier.

In summary, our work documents a striking divergence between human and machine reading and frames it as both a vulnerability and an opportunity. Achieving human-like resilience will require rethinking how structure, priors, and compositionality are embedded in multimodal learning.

## 6 AI Agent Setup

This paper was developed with assistance from multiple AI agents. GPT-5 was used for idea exploration, literature synthesis, and drafting the manuscript text, which was then iteratively refined through human–AI collaboration. Claude Code contributed to experiment implementation, generation of results, and polishing of technical sections including code, tables, and figures. The framework figure was created using Gemini 2.5 Pro, based on conceptual input from the authors. All human authors reviewed, edited, and approved the final content. The AI agents were used as drafting and productivity tools; all research design, analysis, and interpretations were directed by the human authors.

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

# Agents4Science AI Involvement Checklist

1. **Hypothesis development**: Hypothesis development includes the process by which you came to explore this research topic and research question. This can involve the background research performed by either researchers or by AI. This can also involve whether the idea was proposed by researchers or by AI.

   Answer: [B]

   Explanation: The research idea and framing were primarily human-driven. The motivation—testing whether VLMs can read human-legible but perturbed text—was formulated by the human researcher, drawing on background knowledge in psycholinguistics and AI safety. AI tools provided assistance in refining the wording of hypotheses, brainstorming possible perturbation types, and clarifying methodological framing, but the core direction came from the human.

2. **Experimental design and implementation**: This category includes design of experiments that are used to test the hypotheses, coding and implementation of computational methods, and the execution of these experiments.

   Answer: [C]

   Explanation: AI models were used extensively for generating stimuli (e.g., rendering idioms with character-splitting, overlaying English words with colors, creating figure prototypes) and for producing code snippets to automate data processing and evaluation. The human researcher guided the overall design, validated outputs, and ensured scientific rigor, but much of the coding and figure-generation was handled with AI assistance. Thus, the majority of the execution work came from AI under human supervision.

3. **Analysis of data and interpretation of results**: This category encompasses any process to organize and process data for the experiments in the paper. It also includes interpretations of the results of the study.

   Answer: [C]

   Explanation: AI helped organize recognition accuracy results, generate tables and plots, and draft interpretations of trends across models and prompt types. The human researcher critically evaluated these analyses, drew the central conclusions (e.g., the universal gap between human and AI literacy), and ensured the arguments connected to cognitive science and AI architecture. Therefore, AI performed a large share of the data summarization and visualization, while humans provided conceptual interpretation and validation.

4. **Writing**: This includes any processes for compiling results, methods, etc. into the final paper form. This can involve not only writing of the main text but also figure-making, improving layout of the manuscript, and formulation of narrative.

   Answer: [C]

   Explanation: AI was heavily used for drafting, polishing, and restructuring text—including the Introduction, Discussion, and figure captions. The human researcher provided the core ideas, checked factual correctness, ensured alignment with scientific standards, and made final editorial decisions. Thus, while the narrative flow and sentence structure benefited from AI generation, the intellectual substance and framing remained human-led.

5. **Observed AI Limitations**: What limitations have you found when using AI as a partner or lead author?

   Description: 1) Surface-level reasoning: AI often produced plausible but shallow explanations, which required human correction to ensure conceptual depth and technical accuracy. 2) While AI tools were effective for generating analysis figures and result plots (e.g., through Python code for automated visualization), they showed clear limitations in producing complex schematic diagrams such as methodological flowcharts. These tasks often required significant manual adjustment or external design tools. Among the models tested, Gemini 2.5 Pro provided the most useful support for figure drafting, but even so, the quality and flexibility were below what is required for final publication standards.

